# Mutual information deep regularization for semi-supervised segmentation

**Jizong Peng**                    JIZONG.PENG.1@ENS.ETSMTL.CA
**Marco Pedersoli**                  MARCO.PEDERSOLI@ETSMTL.CA
**Christian Desrosiers**             CHRISTIAN.DESROSIERS@ETSMTL.CA
*Ecole de technologie superieure*
*1100 Notre-Dame W., Montreal, Canada (H1C 3K3)*

## Abstract

The scarcity of labeled data often limits the application of deep learning to medical image segmentation. Semi-supervised learning helps overcome this limitation by leveraging unlabeled images to guide the learning process. In this paper, we propose using a clustering loss based on mutual information that explicitly enforces prediction consistency between nearby pixels in unlabeled images, and for random perturbation of these images, while imposing the network to predict the correct labels for annotated images. Since mutual information does not require a strict ordering of clusters in two different cluster assignments, we propose to incorporate another consistency regularization loss which forces the alignment of class probabilities at each pixel of perturbed unlabeled images. We evaluate the method on three challenging publicly-available medical datasets for image segmentation. Experimental results show our method to outperform recently-proposed approaches for semi-supervised and yield a performance comparable to fully-supervised training.

**Keywords:** Semantic segmentation, Semi-supervised learning, Deep clustering, Mutual information, Convolutional neural network

## 1. Introduction

While supervised learning approaches based on deep convolutional neural networks (CNNs) (Long et al., 2015) have achieved outstanding performances in a wide range of segmentation tasks, such approaches typically require a large amount of labeled images for training. In medical imaging applications, obtaining this labeled data is often expensive since annotations must be made by trained clinicians, typically in 3D volumes, and regions to segment can have very low contrast. Semi-supervised learning is a paradigm which reduces the need for fully-annotated data by exploiting the abundance of unlabeled data, i.e. data without expert-annotated ground truth. In contrast to standard approaches that learn exclusively from labeled data, semi-supervised methods also leverage intrinsic properties of unlabeled data (or *priors*) to guide the learning process. Although initially proposed for classification (Oliver et al., 2018), various semi-supervised methods have also been developed for semantic segmentation, including approaches based on self-training (Bai et al., 2017), distillation (Radosavovic et al., 2018), attention learning (Min and Chen, 2018), adversarial learning (Souly et al., 2017; Zhang et al., 2017), entropy minimization (Vu et al., 2019), co-training (Peng et al., 2019; Zhou et al., 2019), temporal ensembling (Perone and Cohen-Adad, 2018),

manifold learning (Baur et al., 2017), and data augmentation (Chaitanya et al., 2019; Zhao et al., 2019a). A simple yet powerful strategy employed in several semi-supervised segmentation methods is transformation consistency (Bortsova et al., 2019). In this semi-supervised strategy, a point-wise loss like Kullback–Leibler (KL) divergence is used to impose similar network outputs for different transformations of the same unlabeled image. Even though this helps make the network robust to such transformations, it does not directly enforce spatial consistency within the image.

Recently, important efforts have been invested toward learning representations from unlabeled data that can be employed as features in a supervised learning task such as classification (Hjelm et al., 2018). A powerful way to obtain such representation is deep clustering (Ji et al., 2018; Caron et al., 2018; Ghasedi Dizaji et al., 2017). However, because clustering is an ill-posed problem, techniques for this task often lead to poor or degenerate solutions (Caron et al., 2018), for instance where all examples are assigned to a single cluster (i.e., mode collapse). To avoid this problem, recent work has proposed using the principle of mutual information (MI) (Weihua Hu, 2017; Zhao et al., 2019b; Ji et al., 2018). The mutual information $I(X, Y)$ between two random variables $X$ and $Y$ is an information-theoretic criterion that measures the dependency between these variables. It is defined as the KL divergence between the joint distribution $p(X, Y)$ of the variables and the product of their marginals:

$$I(X; Y) \;=\; D_{\mathrm{KL}}\big(p(X, Y) \,||\, p(X)\,p(Y)\big). \tag{1}$$

Two significant advantages of MI for clustering, compared to traditional techniques like k-means or Gaussian mixtures, is that it does not make any assumptions about the data distribution and it alleviates the problem of mode collapse by favoring balanced clusters. The second advantage can be seen by an equivalent definition of MI,

$$I(X; Y) \;= H(Y) \,-\, H(Y|X) \tag{2}$$
$$= \; \mathbb{E}_Y \big[ \log \mathbb{E}_X[\, p(Y|X)\,] \big] \,-\, \mathbb{E}_{X,Y}[\, \log p(Y|X)\,], \tag{3}$$

where $H(Y)$ is the entropy of $Y$ and $H(Y|X)$ is the conditional entropy of $Y$ given $X$. If we suppose that $X$ is an image and $Y|X$ is the cluster to which $X$ is assigned then maximizing $I(X; Y)$ can be achieved by increasing the entropy of cluster marginals $H(Y)$, which corresponds to more balanced clusters.

So far, very few works have investigated the usefulness of MI-based deep clustering as a regularization prior for semantic segmentation. In (Zhao et al., 2019b), authors propose a region loss for semantic segmentation which represents a pixel by a patch surrounding this pixel and then maximizes the MI between the distribution of predicted outputs and ground truth labels for this patch. The advantage of this approach over standard segmentation losses like cross-entropy is that it explicitly considers the dependencies between nearby pixels within the loss, thereby enabling spatial regularization. While it achieved better performance than traditional spatial consistency techniques like CRFs (Krähenbühl and Koltun, 2011), this approach only considers fully-supervised segmentation settings. The Invariant Information Clustering (IIC) method proposed for segmentation in (Ji et al., 2018) also considers patches centered on each pixel, however it instead maximizes the MI between the distribution of predicted outputs for a patch and the output distribution for a transformed version of this patch. Two strategies are presented for applying this in a

semi-supervised setting: fine-tuning and overclustering. In fine-tuning, the network is pre-trained on a clustering task using unlabeled images and then fine-tuned on a segmentation task with labeled ones. The second strategy employs unlabeled images to learn a fine-grained clustering and, in a post-processing step, learns a many-to-one mapping from the clusters to segmentation labels based on labeled examples. This mapping uses an algorithm external to gradient descent optimization and labeled images do not participate in the computation of gradients.

**Contributions** In this paper, we propose a semi-supervised segmentation method which leverages both MI-based regularization and transformation consistency in a single model. The major contributions of our work are the following:

- We present a first application of MI for regularization in semi-supervised segmentation, where both labeled and unlabeled images are used simultaneously in an end-to-end manner. The proposed loss function incorporates both fully-supervised guidance from labeled data and an unsupervised regularization term based on MI, which enforces spatial consistency on unlabeled images;

- We extend MI regularization by further encouraging KL-based consistency between the segmentation output for unlabeled images and their transformed version. We show that this additional unsupervised regularization term stabilizes training and leads to higher accuracy;

- We perform an extensive set of experiments on three challenging segmentation benchmarks, comparing our proposed method against recently-proposed approaches for this task. Results show our method to yield significantly higher performance, near to fully-supervised training.

In the next sections, we present our semi-supervised segmentation method and perform experiments demonstrating its advantages over existing approaches.

## 2. Proposed method

Given a labeled set $\mathcal{D}_l$ of image-label pairs $(\mathbf{x}, \mathbf{y})$, with image $\mathbf{x} \in \mathbb{R}^\Omega$, $\Omega = \{1, \ldots, W\} \times \{1, \ldots, H\}$, and ground-truth labels $\mathbf{y} \in \{1, \ldots, C\}^\Omega$ where $C$ is the number of classes, and a large unlabeled dataset $\mathcal{D}_u$ comprised of images without their labels ($|\mathcal{D}_u| \gg |\mathcal{D}_l|$). We want to learn a neural network $f$ parameterized by $\boldsymbol{\theta}$ to predict the label probability distribution of each pixel in an input image. As shown in Figure 1, the proposed model exploits both labeled and unlabeled images during training. Labeled data $\mathcal{D}_l$ is used as in standard supervised methods with a loss $\mathcal{L}_{\text{sup}}$ that imposes the pixel-wise prediction of the network for an annotated image to be similar to the ground truth labels. While other segmentation losses such as Dice loss (Milletari et al., 2016) could have been considered, in this work we employed the well-known cross-entropy loss defined as

$$\mathcal{L}_{\text{sup}}(\boldsymbol{\theta}; \mathcal{D}_l) \;=\; -\frac{1}{|\mathcal{D}_l|\,|\Omega|} \sum_{(\mathbf{x},\mathbf{y}) \in \mathcal{D}_l} \sum_{(i,j) \in \Omega} y_{ij} \log f_{ij}(\mathbf{x}; \boldsymbol{\theta}). \tag{4}$$

In semi-supervised methods, unlabeled data is typically used within a regularization loss to guide the parameter optimization process toward suitable solutions. A popular regularization strategy, called consistency-based regularization, enforces the network to output

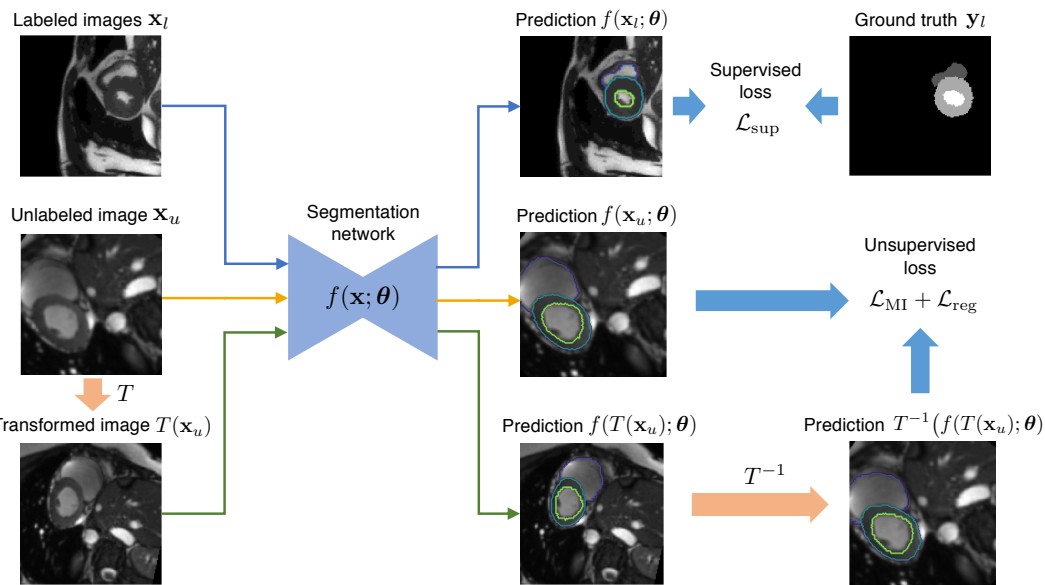

Figure 1: **Training pipeline of our semi-supervised segmentation method**. Given an unlabeled image $\mathbf{x}_u$, a regularization loss is imposed on two related predictions. The first is the prediction of $\mathbf{x}_u$, i.e., $f(\mathbf{x}_u; \boldsymbol{\theta})$, and the second is the prediction of the given image under an invertible transformation $T$, after reversing this transform to return to the original image coordinates, i.e., $T^{-1}\big(f(T(\mathbf{x}_u); \boldsymbol{\theta})\big)$. To highlight the foreground region, images shown here have been center-cropped.

similar predictions for perturbed inputs corresponding to unlabeled data. This strategy is exploited in temporal ensembling techniques like Mean Teacher (Perone and Cohen-Adad, 2018), where the output of a Student network at different training iterations should be similar (e.g., in terms of $L2$ norm or KL divergence) to that of a Teacher network whose parameters are a weighted temporal average of the Student's. A common limitation of such methods is that they regard the prediction for separate pixels as independent in the loss.

**MI-based regularization** To better exploit the structured nature of segmentation, we add a loss term based on MI, denoted as $\mathcal{L}_{\mathrm{MI}}$, which is similar to the one used in (Ji et al., 2018) for deep clustering. In this loss, we represent each pixel $(i, j)$ of an unlabeled image as a patch $\mathbf{p}_{ij} = [\mathbf{x}]_{ij}$ centered on this pixel, where $[\cdot]$ denote a patch extraction operator. The network's output patch $\mathbf{f}_{ij}$ at each position $(i, j)$ can be computed in a single convolution pass using the following relation: $\mathbf{f}_{ij} = f(\mathbf{p}_{ij}; \boldsymbol{\theta}) = [f(\mathbf{x}; \boldsymbol{\theta})]_{ij}$. Considering each output patch $\mathbf{f}_{ij}$ as a distribution, we seek to maximize the MI between this distribution and the one corresponding to adjacent patches. Moreover, we want this spatial consistency to hold for different invertible transformations $T \in \mathcal{T}$ applied to the unlabeled image. We note that loss terms imposing strict equivalence between adjacent patches (e.g., $L_2$ or KL divergence) are not suitable since these patches can be different. In contrast, the MI loss makes a more relaxed assumption that information content does not vary much between adjacent patches, if these patches have a sufficient overlap.

Let $\mathcal{N}$ a be predefined set of pixel displacements $(p, q)$ which defines the neighbors of a pixel $(i, j)$, i.e. $\{(i + p, j + q) \mid (p, q) \in \mathcal{N}\}$. Using the same square patch for all pixels, $|\mathcal{N}|$ then corresponds to the patch size. We define our MI loss as

$$\mathcal{L}_{\mathrm{MI}}(\boldsymbol{\theta}; \mathcal{D}_u) \;=\; \frac{1}{|\mathcal{N}|} \sum_{(p,q) \in \mathcal{N}} I(\mathbf{P}_{pq}), \tag{5}$$

where $I(\mathbf{P}_{pq})$ is the MI given by the joint distribution $\mathbf{P}_{pq}$. Denote as $\mathbf{f}_{ij}^T = T^{-1}\big(f(T(\mathbf{x}_{ij}); \boldsymbol{\theta})\big) = \big[T^{-1}\big(f(T(\mathbf{x}); \boldsymbol{\theta})\big)\big]_{ij}$ the output for a patch transformed by $T \in \mathcal{T}$, after reversing this transform to return to the original patch coordinates. Joint distribution $\mathbf{P}_{pq}$ is a $C \times C$ matrix computed as

$$\mathbf{P}_{pq} \;=\; \frac{1}{|\mathcal{D}_u|\,|\mathcal{T}|\,|\Omega|} \sum_{\mathbf{x} \in \mathcal{D}_u} \sum_{T \in \mathcal{T}} \sum_{(i,j) \in \Omega} \mathbf{f}_{ij} \cdot (\mathbf{f}_{i+p,j+q}^T)^\top \tag{6}$$

Note that the sum in this equation can computed efficiently using a 2D convolution operation. Finally, given the joint distribution $\mathbf{P}_{pq}$, the MI in Eq. (5) is obtained as

$$I(\mathbf{P}) \;=\; \sum_{k=1}^{C} \sum_{k'=1}^{C} \mathbf{P}(k, k') \cdot \log \frac{\mathbf{P}(k, k')}{\big(\sum_{k'} \mathbf{P}(k, k')\big) \cdot \big(\sum_{k} \mathbf{P}(k, k')\big)}. \tag{7}$$

**Transformation consistency** As we will show in experiments, employing $\mathcal{L}_{\mathrm{MI}}$ as the only regularization may however be insufficient to guide the learning towards good solutions. This can be attributed to the clustering nature of the proposed loss. Given two distributions conditionally independent given the same input image, MI is maximized if there is a deterministic mapping between clusters (classes) in each distribution such that they are equivalent. For instance, modifying the cluster indexes in one distribution (e.g., by permutation) does not change the MI. To ensure that learned clusters align across different patch outputs, we add a second regularization term, $\mathcal{L}_{\mathrm{reg}}$, which minimizes the pixel-wise KL divergence between the network output for an image and its transformed version:

$$\mathcal{L}_{\mathrm{reg}}(\boldsymbol{\theta}; \mathcal{D}_u) \;=\; \frac{1}{|\mathcal{D}_u|\,|\mathcal{T}|\,|\Omega|} \sum_{\mathbf{x} \in \mathcal{D}_u} \sum_{T \in \mathcal{T}} \sum_{(i,j) \in \Omega} D_{\mathrm{KL}}\Big(f_{ij}(\mathbf{x}; \boldsymbol{\theta}) \,\big|\big|\, T_{ij}^{-1}\big(f(T(\mathbf{x}); \boldsymbol{\theta})\big)\Big). \tag{8}$$

Our final loss combines the supervised term and the two unsupervised terms based on MI and consistency-based regularization:

$$\mathcal{L}(\boldsymbol{\theta}; \mathcal{D}_l, \mathcal{D}_u) \;=\; \mathcal{L}_{\mathrm{sup}}(\boldsymbol{\theta}; \mathcal{D}_l) \;+\; \lambda\big(\mathcal{L}_{\mathrm{MI}}(\boldsymbol{\theta}; \mathcal{D}_u) + \mathcal{L}_{\mathrm{reg}}(\boldsymbol{\theta}; \mathcal{D}_u)\big), \tag{9}$$

where $\lambda \geq 0$ is a hyper-parameter controlling the relative importance of labeled and unlabeled data.

## 3. Experimental setup

### 3.1. Dataset and metrics

Our experiments are performed on three clinically-relevant benchmark datasets for medical image segmentation: the Automated Cardiac Diagnosis Challenge (ACDC) dataset

([Bernard et al.](#), 2018), the Prostate MR Image Segmentation (PROMISE) 2012 Challenge dataset ([Litjens et al.](#), 2014), and the Spleen sub-task dataset of the Medical Segmentation Decathlon Challenge ([Simpson et al.](#), 2019). The three datasets consist of different image modalities and have various acquisition resolutions.

**ACDC dataset**  The publicly-available ACDC dataset consists of 200 short-axis cine-MRI scans from 100 patients, evenly distributed in 5 subgroups: normal, myocardial infarction, dilated cardiomyopathy, hypertrophic cardiomyopathy, and abnormal right ventricles. Scans correspond to end-diastolic (ED) and end-systolic (ES) phases, and were acquired on 1.5T and 3T systems with resolutions ranging from $0.70 \times 0.70$ mm to $1.92 \times 1.92$ mm in-plane and 5 mm to 10 mm through-plane. Segmentation masks delineate 4 regions of interest: left ventricle endocardium (LV), left ventricle myocardium (Myo), right ventricle endocardium (RV), and background. Short-axis slices within 3D-MRI scans were considered as 2D images, which were re-sized to $256 \times 256$. For our experiments, we used a random split of 8 fully-annotated and 167 unlabeled scans for training, and the remaining 25 scans for validation. We employed conventional data augmentation for both labeled and unlabeled images, including random crop and random rotation within a range of [-20, 20] degrees.

**Prostate dataset**  This dataset is composed of multi-centric transversal T2-weighted MR images from 50 subjects acquired with multiple MRI vendors and different scanning protocols, which are representative of typical MR images acquired in a clinical setting. Image resolution ranges from $15 \times 256 \times 256$ to $54 \times 512 \times 512$ voxels with a spacing ranging from $2 \times 0.27 \times 0.27$ to $4 \times 0.75 \times 0.75$ mm$^3$. We randomly selected 7 patients as labeled data, 33 as unlabeled data, and 10 for validation during the experiments.

**Spleen datset**  This public dataset consists of patients undergoing chemotherapy treatment for liver metastases. A total of 61 portal venous phase CT scans (only 41 were given with ground truth) were included in the dataset with acquisition and reconstruction parameters described in ([Simpson et al.](#), 2019). The ground truth segmentation was generated by a semi-automatic segmentation software and then refined by an expert abdominal radiologist. For our experiments, 2D images are obtained by slicing the high-resolution CT volumes along the axial plane, followed by a max-min normalization with a range between 0 and 1. Each slice is then resized to a resolution of $512 \times 512$. To evaluate algorithms in a semi-supervised setting, we randomly split the dataset into labeled, unlabeled and validation image subsets, comprising CT scans of 4, 32, and 5 patients respectively.

We use the commonly-adopted Dice similarity coefficient (DSC) metric to evaluate segmentation quality. DSC measures the overlap between the predicted labels ($S$) and the corresponding ground truth labels ($G$):

$$\mathrm{DSC}(S, G) = \frac{2|S \cap G|}{|S| + |G|} \tag{10}$$

DSC values range between 0 and 1, a higher value corresponding to a better segmentation. In all results, we report the 3D DSC metric for the validation set.

### 3.2. Implementation details

**Network and parameters**  For all three datasets, we use the same network architecture of U-Net with 15 layers including batch normalization, dropout and ReLU activation. We

adopted this architecture as it has been shown to work well for different medical image segmentation tasks. Networks were trained using stochastic gradient descent (SGD) with an Adam optimizer having a initial learning rate of $1 \times 10^{-3}$ which is decreased during training. To control the relative importance of labeled and unlabeled data in Eq. (9) we used a fixed $\lambda$ of 0.1 for all datasets and experiments. The same strategy was employed to generate transformed images for both $\mathcal{L}_{\mathrm{MI}}$ and $\mathcal{L}_{\mathrm{reg}}$ terms. Given an unlabeled image, we randomly draw a transformation from a pool of invertible transformations, including cascaded transformation of random rotation, shearing and scaling (Xie et al., 2019). The size of patches in the MI loss is also an important hyper-parameter. Patches must be large enough so that information content remains similar between adjacent ones, but small enough to capture local context. In experiments, we used 3×3 pixels which corresponds to regions of 3-5 mm in images depending on the resolution. We also tested our method with 5×5 and 7×7 patches, however this increased computational cost without significantly improving accuracy.

**Comparison baselines** We compared our method against several baselines and recently-proposed approaches for semi-supervised segmentation. First, to get an upper bound on performance, we trained the network described above using the supervised loss $\mathcal{L}_{\mathrm{sup}}$ on *all* training images. We call this baseline Full supervision. Likewise, an lower bound on performance is obtained by optimizing $\mathcal{L}_{\mathrm{sup}}$ only on labeled images, ignoring unlabeled ones. This second baseline is referred to as Partial supervision in our results. Next, we tested two well-known approaches for semi-supervised learning: Entropy minimization (Vu et al., 2019) and Mean Teacher (Perone and Cohen-Adad, 2018). The first approach minimizes the pixel-wise entropy of predictions made for unlabeled images. This forces the network to become more confident about its predictions, and can be seen as a soft version of the pseudo-label algorithm (Vu et al., 2019). For Mean Teacher, we use the same formulation as in (Perone and Cohen-Adad, 2018), where the Student model is trained using labeled data and the Teacher model is updated using an exponential moving average (EMA) of 0.999. The same strategy as (Perone and Cohen-Adad, 2018) is employed to generate transformations for unlabeled data and to impose consistency between Teacher's and Student's predictions for unlabeled images. We report the accuracy of the Teacher, which usually performs better than the Student.

**Ablation study** To assess the impact of our two unsupervised loss terms $\mathcal{L}_{\mathrm{MI}}$ and $\mathcal{L}_{\mathrm{reg}}$ on performance, we performed an ablation study where we disable one of them while keeping the other. Using only $\mathcal{L}_{\mathrm{MI}}$ with the supervised loss $\mathcal{L}_{\mathrm{sup}}$, which we call Mutual information in the results, is similar to the IIC method (Ji et al., 2018) except that in our case MI-based regularization is used *jointly* with the supervised loss in a semi-supervised setting, instead of for pre-training the network on a clustering task before adapting it to segmentation using labeled images. Likewise, using only $\mathcal{L}_{\mathrm{reg}}$ as unsupervised loss with $\mathcal{L}_{\mathrm{sup}}$, which is referred to as Consistency regularization, is similar to the semi-supervised segmentation method recently presented in (Bortsova et al., 2019). Last, following recent work enforcing consistency with $L_2$ distance (Tarvainen and Valpola, 2017), we tested a mean-squared error (MSE) loss instead of KL for $\mathcal{L}_{\mathrm{reg}}$.

Table 1: Mean 3D DSC of tested methods on the ACDC, Prostate and Spleen datasets. RV, Myo and LV refer to the right ventricle, myocardium and right ventricle classes, respectively. We test our method using KL and MSE for $\mathcal{L}_{\mathrm{reg}}$. Mutual information corresponds to our method without loss term $\mathcal{L}_{\mathrm{reg}}$ and Consistency regularization to our KL-based method without $\mathcal{L}_{\mathrm{MI}}$. Reported values are averages (standard deviation in parentheses) for 3 runs with different random seeds.

| | **ACDC** | | | | | |
| | **RV** | **Myo** | **LV** | **Mean** | **Prostate** | **Spleen** |
|---|---|---|---|---|---|---|
| Full supervision | 88.98 (0.09) | 84.95 (0.15) | 92.44 (0.33) | 88.79 (0.13) | 87.33 (0.40) | 93.52 (0.48) |
| Partial supervision | 73.25 (0.36) | 75.54 (1.27) | 86.89 (0.26) | 78.56 (0.42) | 84.20 (0.73) | 87.38 (1.05) |
| Entropy min. | 73.85 (1.29) | 74.92 (0.85) | 86.12 (0.53) | 78.30 (0.87) | 83.04 (0.51) | 90.21 (0.31) |
| Mean Teacher | 82.99 (0.49) | 80.43 (1.02) | 89.33 (0.33) | 84.25 (0.56) | 86.15 (0.19) | 93.22 (0.34) |
| Mutual information | 81.98 (0.62) | 75.75 (0.47) | 87.89 (0.11) | 81.87 (0.32) | 83.75 (1.21) | 90.35 (0.36) |
| Consistency reg. | 82.30 (0.60) | 79.43 (0.81) | 88.55 (0.37) | 83.42 (0.48) | 84.88 (0.54) | 91.50 (0.61) |
| Ours (MSE) | 82.82 (0.35) | 79.91 (0.72) | 88.84 (0.77) | 83.85 (0.39) | 85.77 (0.46) | 93.12 (0.19) |
| Ours (KL) | 85.08 (0.10) | 81.08 (0.42) | 90.72 (0.44) | 85.63 (0.20) | 86.63 (0.07) | 93.37 (0.13) |

## 4. Results

Table 1 reports the mean 3D DSC on the validation set of the ACDC, Prostate (PROMISE) and Spleen datasets. Overall, the proposed method with KL-based loss achieves the highest accuracy for all three datasets. Using a one-sided paired t-test, the improvement of our method over all other approaches is found to be significant ($p < 0.05$) for the RV, LV, Mean of ACDC and the Prostate segmentation tasks. Note that, for the Spleen task, there is no significant difference between our method and the fully-supervised baseline. Improvements are particularly notable for the more challenging task of right ventricle and myocardium segmentation in the ACDC dataset. Furthermore, despite training with a very small fraction of labeled images (i.e., 4.5% of the training set as labeled data for ACDC, 17.5% for Prostate and 3% for Spleen dataset), our method achieves a performance near to full supervision with a DSC difference less than 4% in all cases.

Our ablation study shows that combining both $\mathcal{L}_{\mathrm{MI}}$ and $\mathcal{L}_{\mathrm{reg}}$ regularization losses gives better results than using these losses individually, with statistically significant improvements of 1.65–2.78% compared to using only $\mathcal{L}_{\mathrm{reg}}$ and of 2.83–5.33% with respect to using only $\mathcal{L}_{\mathrm{MI}}$. As expected, employing $\mathcal{L}_{\mathrm{MI}}$ alone yields poor results since $\mathcal{L}_{\mathrm{reg}}$ is also required to align the cluster assignments across different image patches. Comparing KL-based with MSE-based consistency for $\mathcal{L}_{\mathrm{reg}}$, we find the former to give a higher accuracy in all cases. This observation is in line with (Perone and Cohen-Adad, 2018) and recent work on consistency-based unsupervised data augmentation (Xie et al., 2019), showing KL to work well with a wide variety of regularization terms.

The performance of our method can be appreciated visually in Fig. 2, which shows examples of segmentation results for tested methods. One can see that our method better predicts the contour of target regions despite the low contrast in images. On the other hand, using only consistency regularization leads to non-smooth contours of segmented regions.

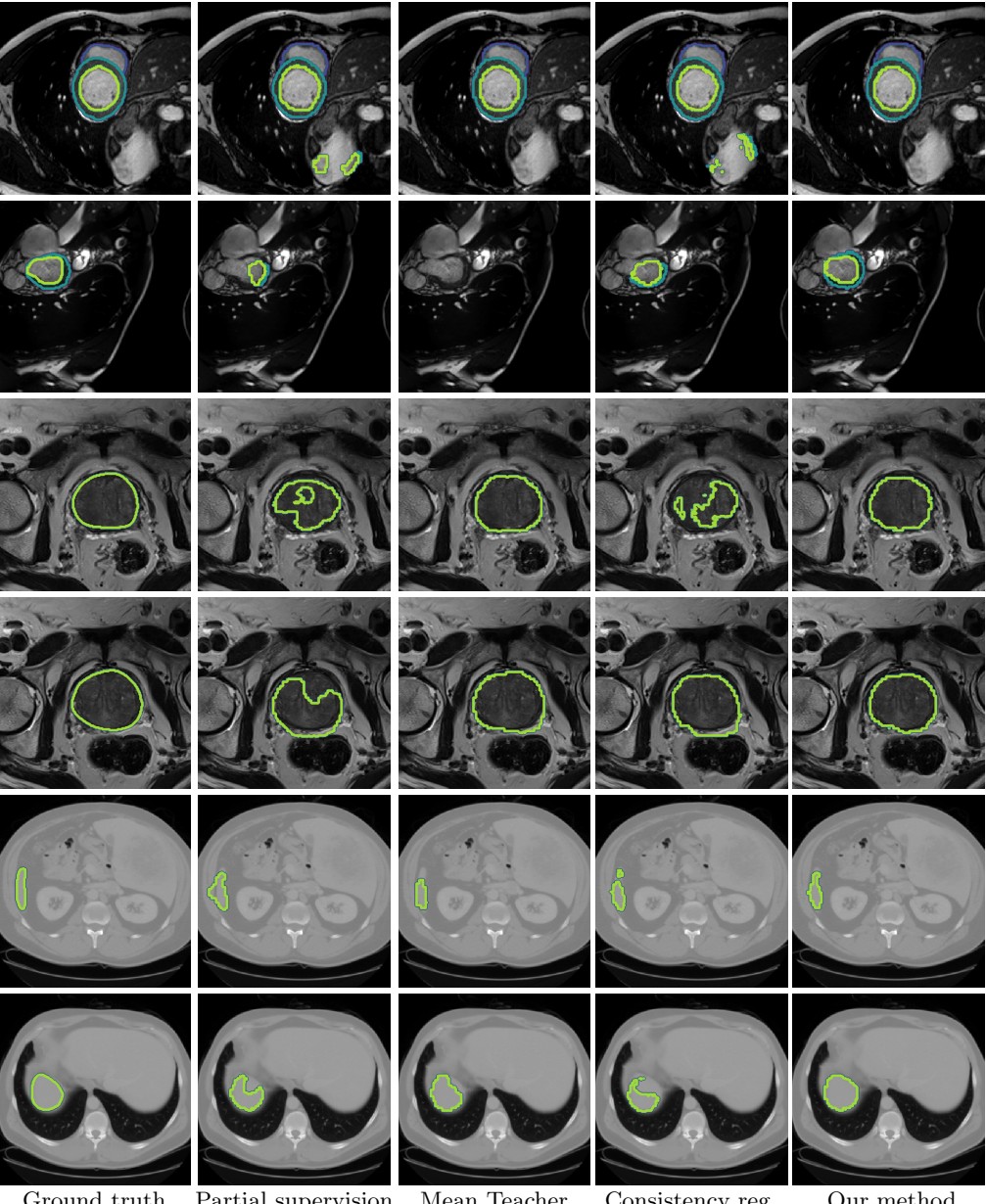

| Ground truth | Partial supervision | Mean Teacher | Consistency reg. | Our method |

Figure 2: Visual comparison of tested methods on validation images. Top two rows: ACDC dataset. Middle two rows: Prostate (PROMISE) dataset. Bottom two rows: Spleen dataset.

## 5. Discussion and conclusion

We presented a novel semi-supervised method for segmenting medical images which regularizes a CNN network for segmentation by maximizing the MI between output distributions for both adjacent patch pairs and images pairs undergoing invertible transformations. Our loss explicitly enforces the network to capture the high-order dependencies between spatially-related pixels, and preserve structure under perturbations on its input. By incorporating the MI within a consistency term, the network can be effectively trained with abundant unlabeled data. We applied the proposed method to three challenging medical segmentation tasks with few images having labeled annotations (4.5% of the training set for ACDC, 17.5% for Prostate and 3% for Spleen). Experimental results showed our method to outperform recently-proposed semi-supervised approaches such as Mean Teacher and Entropy minimization, and to offer an accuracy near to full supervision.

While standard loss function for segmentation consider the prediction for different pixels independently, an important advantage of our MI regularization loss is that it takes into consideration the structured nature of segmentation, where adjacent pixels often have similar class probability distributions. The merit of this loss is demonstrated by the higher DSC score and the more plausible segmentation contours obtained by our method. However, the benefit of MI clustering in semi-supervised segmentation should be further evaluated by providing a deeper theoretical analysis, and validating on large-scale segmentation datasets such as Cityscapes (Cordts et al., 2016). Moreover, due to limited computational resources, we fixed the labeled-unlabeled trade-off hyper-parameter $\lambda$ in Eq. (9) to 0.1 for all three datasets. Likewise, the importance of the two unsupervised losses $\mathcal{L}_{\mathrm{MI}}$ and $\mathcal{L}_{\mathrm{reg}}$ was kept same in all experiments. However, giving more importance to $\mathcal{L}_{\mathrm{MI}}$ could help the network better explore its solution space, as it increases uncertainty in hard-to-segment regions like boundaries. Emphasizing $\mathcal{L}_{\mathrm{MI}}$ could thus potentially alleviate the problem of sub-optimal solutions. Future work could also involve the online optimization of hyper-parameters, for instance based on the concept of hyper-gradient (Baydin et al., 2017), and testing other types of invertible transformations, such as diffeomorphic nonlinear transformations (Narayanan et al., 2005).

## Acknowledgments

We acknowledge the support of the Natural Sciences and Engineering Research Council of Canada (NSERC), and thank NVIDIA corporation for supporting this work through their GPU grant program.

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
