# OpenReview forum: "Mutual information deep regularization for semi-supervised segmentation"
_MIDL.io/2020/Conference — MIDL 2020_

### Official Review · AnonReviewer4 · 2020-03-01
**Mutual information based deep clustering for semi-supervised segmentation**

**Rating:** 4
**Confidence:** 5
**Recommendation:** Oral

**Summary:**

In this paper,  the authors proposed using a clustering loss based on mutual information that explicitly enforces prediction consistency between nearby pixels in unlabeled images, and for random perturbation of these images, while imposing
the network to predict the correct labels for annotated images.
In addition, they proposed to incorporate another consistency regularization loss which forces the alignment of class probabilities at each pixel of perturbed unlabeled images.
Experimental results on several public segmentation datasets demonstrate good performance.

**Strengths:**

1. In the semi-supervised setting for medical image segmentation, supervised loss and unsupervised loss are combined to overcome the requirement for large-scale image annotation.
2. Two unsupervised terms including mutual information and regression loss are derived to further improve the performance.
3. Extensive on the public datasets validate the efficacy of the proposed method.

**Weaknesses:**

1. In the mutual information term, to what large region for the neigbor when forcing the mutual information constraint?
2. For the regression term, are there other transformations to further be explored?
3. A detailed methodological illustration comparing the proposed method to the existing SOTA semi-supervised methods.

**Justification Of Rating:**

The proposed mutual information based deep clusting is interesting under the semi-supervised learning task.
The extensive experiments validated the efficacy of the proposed method.
Overall, this paper is well written and it can contribute to the category of the semi-supervised learning methods.

**Paper Type:**

both

**Special Issue:**

yes

---

> ### Author Response · Authors · 2020-03-28
> **Response to reviewer #4**
>
> We thank the reviewer for appreciating our paper and address comments one by one below.
>
> Comment 1: Size of patch neighborhood.
>
> We fixed the neighborhood size to 3 pixels as it corresponds to anatomically-important differences in medical images (e.g., 3-5 mm depending on the resolution). We also tested neighborhood distances of 5 and 7, however these led to deteriorated performance with increased computational cost. We will make this clearer in the final manuscript.
>
> Comment 2: Other transformations.
>
> In the current implementation, we build transformations by cascading randomly selected operations of rotation, shearing and scaling. However, our model could be used with other types of invertible transformations, for instance, diffeomorphic nonlinear transformations (Narayanan et al. 2005).
>
> Comment 3: Comparison with the state-of-the-art.
>
> We compare our method against two state-of-the-art approaches for semi-supervised segmentation: Consistency regularization and Mean Teacher. Consistency regularization is currently the best performing approach for semi-supervised image classification (Xie et al., 2019; Berthelot et al., 2019), and has shown outstanding performance for semi-supervised segmentation (Bortsova et al., 2019). Likewise, Mean Teacher is considered as one the best methods for semi-supervised learning, and has achieved top accuracy on several semi-supervised segmentation tasks (Cui et al., 2019; Perone et al., 2018) .
>
> References:
>
> Narayanan, R., Fessler, J.A., Park, H. and Meyer, C.R., 2005, July. Diffeomorphic nonlinear transformations: A local parametric approach for image registration. In Biennial International Conference on Information Processing in Medical Imaging (pp. 174-185). Springer, Berlin, Heidelberg.
>
> Xie, Q., Dai, Z., Hovy, E., Luong, M.T. and Le, Q.V., 2019. Unsupervised data augmentation. arXiv preprint arXiv:1904.12848.
>
> Berthelot, D., Carlini, N., Cubuk, E.D., Kurakin, A., Sohn, K., Zhang, H. and Raffel, C., 2019. ReMixMatch: Semi-Supervised Learning with Distribution Alignment and Augmentation Anchoring. arXiv preprint arXiv:1911.09785.
>
> Bortsova, G., Dubost, F., Hogeweg, L., Katramados, I. and de Bruijne, M., 2019, October. Semi-supervised Medical Image Segmentation via Learning Consistency Under Transformations. In International Conference on Medical Image Computing and Computer-Assisted Intervention (pp. 810-818). Springer, Cham.
>
> Cui, W., Liu, Y., Li, Y., Guo, M., Li, Y., Li, X., Wang, T., Zeng, X. and Ye, C., 2019, June. Semi-supervised brain lesion segmentation with an adapted mean teacher model. In International Conference on Information Processing in Medical Imaging (pp. 554-565). Springer, Cham.
>
> Perone, C.S. and Cohen-Adad, J., 2018. Deep semi-supervised segmentation with weight-averaged consistency targets. In Deep Learning in Medical Image Analysis and Multimodal Learning for Clinical Decision Support (pp. 12-19). Springer, Cham.

---

### Official Review · AnonReviewer3 · 2020-03-08
**Well done!**

**Rating:** 4
**Confidence:** 4
**Recommendation:** Oral

**Summary:**

This paper proposed to use mutual information and consistency regularization for semi-supervised learning. Mutual information is conducted on patch level and consistency loss is conducted on pixel level. Experiments are performed using three medical imaging datasets. The experimental results look impressive.

**Strengths:**

1. Good writing, easy to read.
2. The proposed method is sound. Mutual information on the patch level and the consistency loss on the pixel level sounds reasonable to me.
3. Extensive experimental results showed that the proposed method is better than many well known semi-supervised learning methods.


**Weaknesses:**

I have not yet found any significant weaknesses in this paper.
If there is any, I would say it seems to me that the mutual information might be more essential here but the experiments show consistency loss gives better numbers than the MI loss. It would be much appreciated if the authors could provide more insights on this matter.
Furthermore, the results are only reported with a single number each. It is strongly recommended if the authors could run all your experiments 5 times and report the mean and std of the evaluation results.

**Detailed Comments:**

1. Since there were some papers in the CV/ML community which use Norm-2 distance for enforcing the consistency. Maybe, the authors could also include an ablation study where norm-2 distance is used for enforcing the consistency.

2. It would be better if the experiments can be conducted in multiple rounds (with different random seeds) and mean and std values are included in the table.

**Justification Of Rating:**

Semi-supervised learning is an important topic, especially for medical image analysis. This paper is well written. The proposed method is clear and well-motivated. Experiments show strong performance of the proposed method.

**Paper Type:**

methodological development

**Special Issue:**

yes

---

> ### Author Response · Authors · 2020-03-28
> **Response to reviewer #3**
>
> We thank the reviewer for the useful suggestions and address each comment below.
>
> Comment 1: Improvements of Consistency vs MI-based losses.
>
> Many state-of-art approaches for semi-supervised classification are based on transformation consistency, e.g. see (Berthelot et al., 2019), thus it is not surprising that this technique works well for segmentation. As mentioned in our paper, a reason why using the MI loss by itself does not work well is that there is no guarantee that the network will learn the same clusters for different patches, since clusters with permuted labels yield the same MI. Adding the KL term removes this problem by imposing clustering consistency across different patches (via the common pixels in those patches)
>
> Comment 2: L2 norm consistency & multiple runs.
>
> The following table gives the mean and stdev of Dice values obtained by tested methods over 3 runs with different random seeds. The last row of the table is our method using Mean square error (MSE) instead of KL for enforcing consistency.
>
>                                         |        RV          |         Myo      |         LV         |       Mean      |    Prostate   |     Spleen
> Full                                  | 88.98 (0.09) | 84.95 (0.15) | 92.44 (0.33) | 88.79 (0.13) | 87.33 (0.40) | 93.52 (0.48)
> Partial supervision       | 73.25 (0.36) | 75.54 (1.27) | 86.89 (0.26) | 78.56 (0.42) | 84.20 (0.73) | 87.38 (1.05)
> Entropy minimization  | 73.85 (1.29) | 74.92 (0.85) | 86.12 (0.53) | 78.30 (0.87) | 83.04 (0.51) | 90.21 (0.31)
> Mean Teacher               | 82.99 (0.49) | 80.43 (1.02) | 89.33 (0.33) | 84.25 (0.56) | 86.15 (0.19) | 93.22 (0.34)
> Mutual information     | 81.98 (0.62) | 75.75 (0.47) | 87.89 (0.11) | 81.87 (0.32) | 83.75 (1.21) | 90.35 (0.36)
> Consistency reg.           | 82.30 (0.60) | 79.43 (0.81) | 88.55 (0.37) | 83.42 (0.48) | 84.88 (0.54) | 91.50 (0.61)
> Ours (KL)                       | 85.08 (0.10) | 81.08 (0.42) | 90.72 (0.44) | 85.63 (0.20) | 86.63 (0.07) | 93.37 (0.13)
> Ours (MSE)                    | 82.82 (0.35) | 79.91 (0.72) | 88.84 (0.77) | 83.85 (0.39) | 85.77 (0.46) | 93.12 (0.19)
>
> We see that our KL-based loss yields better performance than MSE for enforcing consistency. In a one-sided paired t-test, our method gives significantly higher performance than all other approaches for the RV, LV, Mean (of RV, Myo and LV) and Prostate segmentation tasks. Note that, for the Spleen task, there is no significant difference between our method and the fully-supervised baseline.

---

### Official Review · AnonReviewer2 · 2020-03-13
**Effective method with limited novelty**

**Rating:** 2
**Confidence:** 3

**Summary:**

In this paper, the authors propose a semi-supervised segmentation method using a combination of cross-entropy loss for supervised training and mutual information loss plus consistency loss for unsupervised training. The method is validated on three different datasets and proved to be effective in semi-supervised segmentation tasks. However, the technical novelty is limited.

**Strengths:**

(1)  This paper is trying to solve an important problem in medical image segmentation tasks, i.e., how to make full use of unlabeled data to promote the performance of the model. And give a feasible approach to tackle this problem.
(2) The paper is well written and easy to follow.
(3) The proposed method is validated on three different datasets and achieves good performance.

**Weaknesses:**

(1) The novelty is limited. This method simply combines several loss functions to solve a semi-supervised segmentation problem, i.e., the cross-entropy loss for labeled data, the mutual information loss from ICC method and consistency loss from (Bortsova et al., 2019) for unlabeled data.
(2) The author claims that the IIC method (Ji et al., 2018) is used to pre-train a segmentation network and needs to be fine-tuned on labeled images. But I didn't find such descriptions in the IIC's paper. And I think IIC can provide segmentation results without fine-tuning on labeled images. Just need to find the correspondence between the clustering results and the ground-truth labels.

**Justification Of Rating:**

Although the method in this paper is not novel, it indeed solves an important problem and achieves good performance on three different datasets. I'd like to accept this paper if the authors can address all my concerns.

**Paper Type:**

methodological development

**Questions To Address In The Rebuttal:**

(1) Explanation about point (2) in weakness.
(2) How to split the data of each dataset into labeled, unlabeled and validation sets? Randomly or not?
(3) The paper mainly focuses on mutual information loss, but the results using consistency loss are better than only using mutual information loss (Table 1). Any explanation about this phenomenon?

**Special Issue:**

no

---

> ### Author Response · Authors · 2020-03-28
> **Response to reviewer #2**
>
> We thank the reviewer for highlighting the good performance of our method, and address each comment below.
>
> Comment 1: Limited novelty.
>
> The paper by Bortsova et al. was submitted to Arxiv on November 11th 2019, and we became aware of this paper only at the time of writing. Nevertheless, there are notable differences between the consistency loss employed in our work and theirs. First, while we employ KL as consistency loss, they instead use soft-IoU (we plan to test this loss function in future work). Second, their approach requires a differentiable transformation (e.g., elastic deformation in their paper) whereas our model does not have this limitation. Third, in our work, transformation consistency is not only employed for regularization but also to help align the clusters of the MI-based loss with segmentation labels during training. As shown in our experiments, combining the MI-based and consistency losses helps boost the performance of these individual techniques.
>
> Comment 2: Semi-supervised IIC.
>
> We thank the reviewer for this observation. In (Ji et al., 2019), two strategies are proposed for extending IIC to a semi-supervised setting: fine-tuning and overclustering. In fine-tuning, the network is pre-trained on a clustering task using unlabeled images, and then fine-tuned on a segmentation task using labeled ones. In the overclustering strategy, the network first uses unlabeled images to learn a fine-grained clustering with more clusters than actual segmentation labels. In a post-processing step, labeled examples in the training set are then used to learn a many-to-one mapping from clusters to ground-truth labels. This mapping is found with an algorithm external to the gradient descent optimization, hence labeled images do not participate in the computation of gradients. The post-processing task of finding the optimal mapping can be seen as a weaker form of supervised adaptation, compared to the fine-tuning approach, where modifications to the network are limited to the final cluster-to-label assignments. Unlike these two strategies, our model leverages both labeled and unlabeled images *jointly* during training.
>
> Comment 3: Data split.
>
> As described in the paper, for ACDC dataset, we randomly choose 8 scans as labeled data, 167 scans as unlabeled data, with the remaining as validation data. For the Prostate and Spleen datasets, the splits are randomly chosen as well.
>
> Comment 4: Improvements of Consistency vs MI-based losses.
>
> Many state-of-art approaches for semi-supervised classification are based on transformation consistency, e.g. see (Berthelot et al., 2019), thus it is not surprising that this technique works well for segmentation. As mentioned in our paper, a reason why using the MI loss by itself does not work well is that there is no guarantee that the network will learn the same clusters for different patches, since clusters with permuted labels yield the same MI. Adding the KL term removes this problem by imposing clustering consistency across different patches (via the common pixels in those patches).
>
> References:
>
> Ji, X., Henriques, J.F. and Vedaldi, A., 2019. Invariant information clustering for unsupervised image classification and segmentation. In Proceedings of the IEEE International Conference on Computer Vision (pp. 9865-9874).
>
> Berthelot, D., Carlini, N., Cubuk, E.D., Kurakin, A., Sohn, K., Zhang, H. and Raffel, C., 2019. ReMixMatch: Semi-Supervised Learning with Distribution Alignment and Augmentation Anchoring. arXiv preprint arXiv:1911.09785.

---

> > ### Comment · AnonReviewer2 · 2020-04-03
> > **Address most of my concerns**
> >
> > The rebuttals address most of my concerns, therefore, I will change my rating to weakly accept.
> > I have one more question. Did you compare the results of your method to the results of finetuning on an unsupervised model using IIC method? I'd like to know whether the proposed joint learning strategy outperforms the fine-tuning strategy.

---

### Official Review · AnonReviewer1 · 2020-03-18
**interesting semi-supervised learning approach with good experiments and results however with a misleading title.**

**Rating:** 2
**Confidence:** 4

**Summary:**

The authors propose an image registration based method to perform semi-supervised learning of medical images. The method involves registering an unlabelled image to a label image, passing it through segmentation network to produce segmentations that has maximum mutual information with the segmentation of the labelled image. Experiments show that the performance of this method using a fraction of the data reaches dice scores close to the dice scores of networks trained with the entire dataset.

**Strengths:**

1. The paper is an easy read, however requires a check for grammar at several places..
2. The methodology is interesting and simple.
3. The experimental setup is well thought and the results are good.

**Weaknesses:**

1. The use of mutual information to compare transformed segmentations is unclear. Why not use dice scores instead?
2. There a couple of works that use image registration to perform semi-supervised, single shot learning (Chaitanya et al IPMI, Dalca, et al CVPR).
3. The use mutual information for semi-supervised learning is not novel as claimed in the paper.
4. Where is the clustering part? The title is quite misleading.
5. MI and KL are intimately related. If you maximise one, the other is minimised. However in the cost function they have the same sign. In addition, one can just add a factor of 2 to MI term to get a similar effect. Thus the impressive improvement of dice when compared to mutual information experiment is unclear.

**Justification Of Rating:**

While the idea of the paper is novel, the methodology seems very ad-hoc and not very well thought. The results are good but counter intuitive and need better explanation. A better study of current literature in this space is also essential.

**Paper Type:**

methodological development

**Special Issue:**

no

---

> ### Author Response · Authors · 2020-03-28
> **Response to reviewer #1**
>
> We thank the reviewer for the useful comments which we answer one by one below.
>
> Comment 1:  Mutual-information (MI) to compare transformed segmentations.
>
> Dice (or KL) is a pointwise loss which imposes having similar predictions for corresponding pixels. In our MI-based loss, we want to enforce information invariance locally between *surrounding neighbor patches* in transformed images. Because compared patches are *different, a loss imposing strict equivalence like Dice would not work. However, such a loss could be useful to compare the same patch in different transformed images (after reversing the transformation), and this is why we use KL in our regularization loss Lreg.
>
> Comment 2:  Additional references.
>
> We thank the reviewer for the two suggested references which we will add in the final manuscript. Although related, these works are however very different from ours. The paper by Chaitanya et al. focuses on unsupervised (task-driven) data augmentation, and does not make use of MI, nor concepts of information invariance. Note that our proposed model is transformation-agnostic (as long as the transformations are invertible) and could easily accommodate this type of transformation. Furthermore, the work by Dalca et al. only uses MI to align multimodal images (T1 and FLAIR) in pre-processing. Additionally, it requires a large set of labeled images to learn the anatomical prior, and therefore is not applicable to our semi-supervised scenario.
>
> Comment 3:  Novelty of the proposed method.
>
> In the literature on image analysis, MI has mostly been used for image registration, clustering and representation learning. To our knowledge, the IIC method in (Jie et al., 2019) is the first one using MI for segmentation in a reduced supervision setting. However, IIC is proposed as an unsupervised method, and its semi-supervised extension does not use labeled and unlabeled data jointly in training, as in standard semi-supervised approaches. Specifically, two strategies are proposed for extending IIC to semi-supervised learning. 1) fine-tuning: IIC is used as a clustering method on unlabeled data to learn a representation and the representation network is fine-tuned for segmentation on labeled data. 2) overclustering: using unlabeled images, an overclustering of instances (pixels or images) is learned and, afterwards, labeled training images are used to find an optimal mapping between the many clusters to actual labels. Unlike these two strategies, our method i) uses labeled and unlabeled images simultaneously during training and ii) enforces an explicit alignment of learned clusters and labels with a separate loss term (i.e., Lreg).
>
> Comment 4: Use of clustering in the title.
>
> We used the term clustering in the paper since our method uses MI as a way to ensure consistency between spatially-related pixels. In semi-supervised learning, regularization priors like Laplacian regularization are sometimes referred to as clustering priors. Nevertheless, we agree that a term like “regularization” would be more appropriate than “clustering” and, if possible, we will make this change in the final manuscript.
>
> Comment 5: Relationship between KL and MI.
>
> We thank the reviewer for this comment. MI and KL are indeed related however their relationship is not as straightforward. For instance, two discrete distributions can have both high MI and KL values (e.g., a distribution P and one obtained by permuting the labels of P). In this sense, MI is less restrictive than KL and only enforces information invariance. In our model, the MI and KL loss terms are used in different ways: Mi is employed between different patches that are near each other, whereas KL is only used between the same patches in different transformed images. Therefore, we cannot directly compare the improvements obtained individually by these two terms.
>
> References:
>
> Ji, X., Henriques, J.F. and Vedaldi, A., 2019. Invariant information clustering for unsupervised image classification and segmentation. In Proceedings of the IEEE International Conference on Computer Vision (pp. 9865-9874).

---

> > ### Comment · AnonReviewer1 · 2020-03-28
> > **Unconvincing response.**
> >
> > I would like to thank the authors for their response. However, I find them unconvincing for several reasons. A few are below:
> >
> > 1. "Data augmentation using learned transformations for one-shot medical image segmentation" this was the Dalca paper referred to, not the image registration paper.
> > 2. MI is used in this paper for evaluating pairwise alignment. It is not used to cluster anything in this algorithm. The term is simply misleading.
> > 3. Regarding Jie et al., as very clearly mentioned in the paper, MI is used to find a mapping that maximise MI between ALL possible pairs of data iteratively for clustering. In this paper, there is no clustering unfortunately, it is a simple pairwise measure between f(I) and a f(phi. I). Even ignoring this, Jie et al, use IIC to find "good representations" which are then fine tuned in a supervised setting which is quite different to the objective of the proposed method.
> > 4. This work reminds me a lot of enforcing consistency, an example is here -- https://arxiv.org/pdf/1908.05959.pdf. The ONLY difference as I see is the regularisation term. Is that correctly understood?
> > 5. Regarding MI and KL, as I understand it, MI is computed between local patches and KL is computed for the entire image. Why not compute MI for the entire image?
> > 6. The MI information is computed between the posterior probabilities given by the network. This does not make sense since there is no LOCAL information to extract like in the images. this really needs to be clarified.
> >
> > Overall, I still stand by my previous review that this work is of very, very limited novelty. Especially the work mentioned in 4) seems very, very similar.

---

> > > ### Author Response · Authors · 2020-04-02
> > > **Additional clarifications**
> > >
> > > We thank the reviewer for explaining his/her decision, and take the opportunity to further answer comments.
> > >
> > > 1. We assume the reviewer is referring to the paper by Zhao et al. titled “Data augmentation using learned transformations for one-shot medical image segmentation”. Although interesting, this paper is also very different from our work. It proposes to generate synthetic images and ground truth by learning to transform an atlas (i.e., warping and intensity transform) to match training samples. Mutual information or concepts of information invariance are not used in the paper.
> > >
> > > 2. Unsupervised segmentation (where we have no definition of segmentation labels) is essentially a pixel-level clustering problem where pixels in an image are grouped based on their spatial proximity and feature similarity, e.g. see (Shi and Malik, 2000). In semi-supervised segmentation, where some images have annotations but most images don’t, clustering priors on pixels are used as a regularization technique to enforce spatial smoothness in predicted segmentations. Therefore, while we maintain that our work is based on the concept of clustering, we also agree that the term “spatial regularization” would be more appropriate in the title and that the link between clustering and our method should be clarified in the paper.
> > >
> > > 3. As mentioned in our previous answer, we use MI to enforce spatial smoothness in the segmentation by imposing the prediction for patches of neighbor pixels to have a high mutual information. The advantages of MI compared to KL for this task are twofold. First, patches around two neighbor pixels can be different (especially for large neighborhood sizes) and thus we cannot impose the prediction for these patches to be the same (i.e., same labels at the same position). However, it is much more reasonable to assume that the information content will not vary much between these patches, if they have a significant overlap. Second, suppose that we have no labeled images (unsupervised segmentation), then using only KL to regularize the output (i.e., using Lreg alone) would lead to trivial solutions where the same label is predicted for every pixel (thus giving a KL of 0). Because it favors balanced clusters, MI doesn’t suffer from this problem.
> > >
> > > 4. Transformation consistency, which draws from the well-known principle of data augmentation, is used in many recent papers. Our work does not propose transformation consistency by itself as a novel contribution, but instead uses it as a way to improve the MI-based regularization in a semi-supervised setting.
> > >
> > > Except for the fact that it also uses transformation consistency, the paper by Orbes-Arteaga et al. mentioned by the reviewer is quite different. It focuses on multi-domain adaptation, not semi-supervised segmentation, and proposes an adversarial model for training.
> > >
> > > 5. See our answer to Comment #3 for the difference between MI and KL. Regarding the computation of MI over the entire image, this is certainly possible however it would not be very useful because information on locality would be lost in the process. This is because the joint probability distribution matrix P aggregates probabilities over every pixel in the region of interest. By instead considering smaller patches, the MI-based loss imposes information invariance on a more local scale.
> > >
> > > 6. As mentioned in the previous answer, the MI is computed by aggregating output probabilities over all pixels in local patches. Hence, information invariance is enforced locally (i.e., over a small region and between nearby pixels).

---

### Meta-Review · Area_Chair1 · 2020-04-07
**MetaReview of Paper99 by AreaChair1**

**Rating:** 3
**Recommendation For Accepted Papers:** Poster

**Metareview:**

The reviewers found the paper interesting, and overall there is probably enough support to accept the paper into MIDL. I also appreciate the thorough and clear answers by the authors. However, the reviewers noted very important issues that **need to be corrected** in the camera ready.

Reviewer #1's comments about existing literature on semi-supervised segmentation was brushed away by the authors, which I find quite troubling. Papers mentioned by Reviewer #1, which might appear as "data augmentation" papers in the title and use a completely different methodology (not using MI), definitely tackle the same problem (semi-supervised segmentation) where they take advantage of very limited labelled data and a host of unlabelled data. Including a discussion of related semi-supervised segmentation work and placing the work in that context is important for the paper to be complete. Similarly, adding the Bortsova paper in the introduction is a requirement -- it did not only appear on arxiv at the end of the year, but rather was properly published at MICCAI 2019. The updated results with variance estimations also need to be in the paper.

I also encourage the authors to also take all the other feedback and incorporate it in the paper.



**Paper Type:**

methodological development

**Special Issue:**

no

---

> ### Author Response · Authors · 2020-04-07
> **Thank you for the meta-review**
>
> We thank the AC for the detailed meta-review and helpful suggestions that we will implement in the camera-ready paper. To answer the AC's comment about existing literature, we believe that Reviewer#1's comments were not brushed away in our rebuttal. In our response, we agreed to add the suggested references to the final manuscript and were simply pointing out the important differences between these previous works and our paper. These differences will be clearly discussed in the camera-ready manuscript. We will also incorporate all other feedback while updating the paper.

---

> > ### Comment · Area_Chair1 · 2020-04-07
> > **references**
> >
> > Thank you for the response.
> >
> > Reading the second response to R1 ("We assume the reviewer..."), I did not get the feeling that you intended to discuss the papers in the paper, but perhaps it was my reading of the comment. I'm happy to hear you will take all the feedback and references to the camera ready.

---

### Decision · Program_Chairs · 2020-04-11

Accept